# Population travel increases the risk of *Plasmodium falciparum* infection in the highland population of Gardula Zone, South Ethiopia: A longitudinal study

Muluken Assefa[1], Fekadu Massebo[2], Temesgen Ashine[1], Teklu Wegayehu[2]*

1 Malaria and Neglected Tropical Diseases, Armauer Hansen Research Institute, Addis Ababa Ethiopia,
2 Department of Biology, College of Natural Sciences, Arba Minch University, Arba Minch, Ethiopia

* tekluweg2007@yahoo.com

## Abstract

Population movement influences malaria epidemiology and can be a threat to malaria control and elimination. In Ethiopia, highland dwellers often travel to lowland areas where malaria is endemic. The current study aimed to assess the incidence of malaria and risk factors among dwellers in two highland villages of the former Dirashe District (now Gardula Zone), South Ethiopia. A longitudinal study was conducted from 10/05/2018 to 30/11/2018. A total of 1672 individuals from 329 households were recruited via a systematic random sampling technique. Blood samples were collected from all consented family members. The study participants were interviewed via a pretested questionnaire. Bivariate and multivariate analyses were conducted to determine risk factors associated with malaria infection. A total of 4,884 blood samples were screened for malaria parasites in three rounds of surveys. Among those, 82 slides were positive for malaria parasites, 70 (85%) of which were collected during active case detection, and the remaining 12 (15%) were captured by passive case detection. *Plasmodium falciparum* accounted for 69.5% (57), and the remaining 30.5% (25) were *Plasmodium vivax*. The incidence of malaria in the highlands of Dirashe District was 0.2 infections per person-year at risk. Inhabitants who traveled to lowlands in the past 30 days (AOR = 2.60, 95% CI: 1.27–5.33) had a significantly greater risk of contracting *Plasmodium falciparum* infection. Those people traveling in May and November, those participants who had no formal education and agricultural workers, had a greater risk of developing malaria infection. Low bed net ownership (63.2%) and use (52.9%) have been documented among highland populations. This study revealed that people who travel from highlands to malaria-endemic lowland areas for agriculture are at increased risk of developing malaria. Hence, malaria interventions targeting travelers should be implemented to reduce the imported malaria burden in highlands.

**Data Availability Statement:** All data for the conclusion are included in the manuscript.

**Funding:** This study was financially supported by Arba Minch University and NORHED.

**Competing interests:** The authors have declared that no competing interests exist.

## Introduction

Malaria is a major public health problem worldwide. The African region of the World Health Organization (WHO) has disproportionally the highest disease burden, accounting for 94% of cases and deaths [1]. In Ethiopia, malaria is a leading vector-borne disease and is among the top ten causes of morbidity and mortality [2]. According to Ethiopian Public Health Institute, more than half of the country's landmass is malarious, and 60% of the population lives in this area [3]. The epidemiological pattern of malaria transmission is mostly unstable and seasonal, with varying intensities across ecoepidemiological settings [4, 5].

Malaria transmission intensity is high in areas below 2000 m above sea level (masl). However, highlands and fringe areas in Ethiopia have extensive temporal records of malaria transmission caused by both *Plasmodium falciparum* and *P. vivax* [6–8]. Climatic change may promote the expansion of malaria to highlands [7, 8]. Importantly, increasing temperature may hasten the development of immature stages of malaria vectors and increase the human biting rate of adult mosquitoes [9–12]. Moreover, increasing temperature decreases the duration of the gonotrophic cycle and extrinsic incubation period of *Plasmodium* parasites [10, 11].

People living in highlands and their fringes might lack prior exposure to *Plasmodium* parasites and associated immunity against malaria infection [13–16]. In such settings, all age groups are susceptible to malaria infection upon infection with infectious mosquitoes. Thus, the movement of nonimmuned highland populations to lowland areas where malaria is endemic might increase the risk of malaria infection in highland residents [17, 18]. Consequently, the geographical expansion of malaria to highlands could be a serious public health concern, as most people are living in the highlands of Ethiopia.

Several *Anopheles* species, namely, *Anopheles arabiensis*, *An. christyi*, *An. cinereus* and *An. demeilloni* have been documented in highland areas in Ethiopia [19–21]. *Anopheles christyi* and *Plasmodium* circum-sporozoite protein-positive *An. demeilloni* have been documented in the highlands of the former Dirashe District (now Gardula Zone) [22]. Few studies have attempted to characterize the entomological and parasitological parameters of malaria transmission in the highlands of the study area [22, 23]. However, evidence on malaria transmission has not been well established and is not adequate for determining whether transmission is indigenous or imported.

Although the majority of the Dirashe population lives in nonmalarious highlands, their agricultural lands are located in malaria-endemic lowland areas. Thus, the existence of farmlands in malaria-endemic lowland areas has forced highland dwellers to travel to malarious lowlands for agriculture and other daily activities. However, evidence on the risk of malaria infection associated with population movement is limited in this area. Hence, there is a need to better understand malaria transmission patterns in the highlands of Ethiopia, which can help in planning and implementing appropriate control interventions. Therefore, the present study aimed to assess the incidence of malaria and associated risk factors among dwellers in the highlands of Gardula Zone, South Ethiopia.

## Materials and methods

### Study setting

This study was conducted in two villages (Walyte and Layignaw-Arguba) in the former Dirashe District (now Gardula Zone), South Ethiopia. The total area of the district is 699.383 square km, and the total population is estimated to be 144,593 (85,339 males and 59,254 females) (Unpublished District health office data, 2016). The altitude of the district ranges from 1140 to 2614 masl. The average altitude of Walyte is 2100 masl, and it is 2337 masl in

Layignaw-Arguba. The people receive primary health care services from one primary hospital, four health centers, sixteen health posts and thirty-four private health facilities. The villages were selected on the basis of the number of malaria cases observed at health facilities and populations traveling to the lowlands for agricultural work.

The annual rainfall ranges between 600 and 1600 mm, and the annual temperature between 14.1 and 27.5˚C. The district has two agricultural seasons, which occur simultaneously with the rainy season (the bimodal rainy pattern). Farming of cereal crops, especially sorghum, maize and *teff*, as well as cash crops, such as khat (*Catha edulis forskal*), is the main mode of living. The source population includes all the inhabitants of Gardula Zone, South Ethiopia, whereas the inhabitants of Walyte and Layignaw-Arguba villages of the zone constitute the sample population.

## Study design

A longitudinal study with active and passive case detection was conducted from May 2018 to November 2018. The sampling periods were chosen to coincide with a peak malaria transmission season following the rainy season and the dry season to measure malaria infection. The three months, May, July and November of the year, are characterized by high population movement to malarious lowlands for farming, either for land preparation, cultivation, or harvesting, and hence are selected purposively.

## Active case detection

Active case detection (ACD) was conducted in three rounds, each lasting, on average, one month. To minimize loss to follow-up, the recruited households were sensitized by volunteer health workers prior to the date of the survey. The included households were visited by trained laboratory technicians to screen all family members for malaria parasites via the Rapid Diagnostic Test (RDT). Back tracing for lost participants at the time of the household visit was performed the next morning to ensure maximum participation.

## Passive case detection

During the ACD period, participants who had fever were advised to visit nearby health care facilities for illnesses, including malaria. Three government health care facilities were selected for passive case detection (PCD) on the basis of their accessibility for the study villages and availability of care for malaria. A unique card was prepared and provided for easy identification of the participants. Moreover, a preprepared registration format that contains the list of all household members was made available in each selected health care facility. Those who presented to health care facilities for malaria treatment and were confirmed by microscopy were considered. The test results were recorded by a trained laboratory technician according to the identification card in the registration format prepared for the study. The test results were collected by the principal investigator from the registration format in the selected health care facilities.

## Sample size determination and sampling

A single population proportion formula was used to calculate the sample size (shown below) with a 95% confidence interval, 2.5% margin of error, 50% anticipated malaria prevalence and 5% loss to follow-up, and the single population proportion formula was used to calculate the sample size. A total of 1,612 individuals were included in the study. To minimize the loss to follow-up and for easy access of the study participants, the calculated sample size was changed

to households by dividing it by 4.9, the average family size of the population for the former South Nations, Nationalities and People's Regional states [24]. Finally, 329 households were recruited for the study.

$$n = \frac{(Z1-\frac{\alpha}{2})^2 * P * (1-P)}{d^2}$$

where n = the sample size.

$Z_1$-α/2 = the Z value at a given confidence level

P = estimated prevalence of malaria in the study population

d = margin of error or sample error

Accordingly,

$$n = \frac{(1.96)^2 * 0.50 * (1-0.50)}{0.025^2} = 1536 + 5\% = 1612 n = \frac{(1.96)^2 * 0.50 * (1-0.50)}{0.025^2} = 1536 + 5\% = 1612$$

We used the lists of households of the two selected villages as a sampling frame. Similar exposure of the household inhabitants was assumed to select the households. Therefore, a systematic random sampling technique was employed to include the 329 households. The sample size was allocated proportionally to each village on the basis of their total number of households. Hence, the number of households needed for the survey in Walyte was 171, and the number needed for the survey in Layignaw-Arguba was 158. The eligible households were identified from a list in each village health post and again cross-checked by visiting each household. The first household was selected randomly via the lottery method, and every 9th household was included in the study. If the household's occupants refused to participate in the survey, the household nearest to it was selected.

## Blood sample collection and processing

Blood samples were collected following the standard finger-prick method [25] for malaria testing via RDT and microscopy. The RDT used was CareStart™ Malaria, which detects *P. falciparum* and *P. vivax* species of malaria with high accuracy. After informed consent was obtained from adults and assent from parents/guardians for children, a fingerprick was performed via a sterile disposable lancet by trained laboratory technicians from all inhabitants of selected households during house-to-house visits. For participants who were positive for *Plasmodium* parasites with RDT, thick and thin blood smears were prepared following standard procedures as detailed by the World Health Organization [25]. The slides were transported to the Biology Department Laboratory, Arba Minch University, with a slide box for confirmatory reading.

## Risk factor assessment

At the time of ACD, information was collected via trained data collectors via a pretested and structured questionnaire covering sociodemographic and individual risk factors for malaria infection, including recent travel history to an endemic area. It was first prepared in English and then translated into the Amharic language and Dirashegna (a local language) for easy communication and then translated back to English by language experts to check its consistency. To minimize recall bias, an Ethiopian calendar was used to help participants recall their travel history to endemic areas. When the names of the villages where the participants traveled were recorded, the data collector used a list of village names provided by the former Dirashe district. Each participant was first asked about any overnight stays away from their residence village in the past 30 days and then asked to identify when and where they had traveled and used long-lasting insecticide-treated nets (LLINs) during travel.

Participants who visited health care facilities for malaria treatment during the study period were interviewed with the same questions, and the data were captured by data collectors from selected health care facilities. Households were surveyed in September 2018 to assess coverage

and usage of LLINs via a pretested and structured questionnaire covering malaria prevention and control strategies.

## Data quality assurance

To ensure the quality of the data, training was given to the data collectors on the objective of the study, data collection, respondents' approach, and confidentiality before the data collection. The collected data were checked for consistency and completeness daily. To assure the laboratory result, microscopic confirmation of all slides was conducted via a senior laboratory technologist who was blinded to the RDT results. Slides reported as positive for either *P. falciparum* or *P. vivax* were considered malarial infections. The second-round examination was conducted for slides reported as negative in the first round of examination. The second reader was also blinded to the RDT and first-round microscopy results. The *Plasmodium*-negative slides identified by the second readers were considered negative, and the positive slides were considered positive.

## Data management and analysis

The data were entered and cleaned via a Microsoft Excel spreadsheet and exported to SPSS version 20.0 software for analysis. The malaria incidence rate was calculated as the total number of cases within the study period per person-year at risk. The incidence rate was stratified into three age groups: 1–4 years, 5–14 years, and ≥15 years. The distribution of incidence rates among age groups was analyzed via the chi-square test.

Risk factors for malaria infection were analyzed via a multivariate logistic regression model. The independent variables were sex, age, education, occupation, survey month, village, travel history in the past 30 days, and LLIN usage during travel. The bivariate logistic regression method was used to analyze the associations between the independent variables and the dependent variable (malaria infection status). Variables found to have an association with the dependent variable with a p value less than 0.2 in the bivariable analysis were entered into the multivariable logistic regression model to control for the possible effects of confounders. The variables that had significant associations were identified on the basis of adjusted odds ratios (AORs), 95% CIs and p values <0.05.

The coverage and utilization of the LLIN data were analyzed via descriptive statistics. Household LLIN ownership was calculated as the percentage of households that own at least one LLIN. Household LLIN usage was calculated as the percentage of participants who slept under LLINs the night before the visit.

## Ethical considerations

Ethical clearance was obtained from the Institutional Review Board of the College of Medicine and Health Sciences, Arba Minch University (Ref. No. CMHS/120113/111). Permission letters were obtained from the district health office and the village administrative services by explaining the purpose of the study. The benefits and risks of participating in this study were explained to the participants. Informed consent for participants under 18 years of age was obtained from the heads of household/guardians, and written informed consent was obtained from participants above 18 years of age. Participants positive for malaria with RDT or microscopy were treated according to the Ethiopian National Malaria Treatment Guideline in nearby public health facilities free of charge. Participant records were coded on each respective questionnaire and accessed only by the research team members to maintain confidentiality.

## Results

### Sociodemographic characteristics

A total of 1,672 individuals from 329 households were recruited for this study to measure malaria incidence. The loss to follow-up rate on the last survey conducted in November 2018 was 7.9% (132). Most of the participants (62.8%) were 15 years old or older, with a mean age of 22.8 years. The numbers of males and females were almost equal. The major livelihood (46.6%) of the study participants was agriculture (Table 1).

### Malaria incidence

Out of the 4,884 blood samples tested in three-month surveys, 82 malaria cases were confirmed by microscopy. The majority of cases, 85% (70), were identified during ACD, and the remaining 15% (12) were captured by PCD. *Plasmodium falciparum* accounted for 69.5% (57), followed by *P. vivax at* 30.5% (25).

Overall, the incidence of malaria infection in the study area was 0.2 infections per person-year at risk (Table 2). The incidence of malaria infection varied across survey months; sex, occupation and locality of residence (village) of the study participants. The incidence of *P. falciparum* malaria infection was significantly greater in May (OR = 3.14, 95% CI: 1.48–6.69, p < 0.05) and November (OR = 2.43, 95% CI: 1.10–5.35, P < 0.05) than in July. *Plasmodium falciparum* infection was also more common among male participants (OR = 2.05, 95% CI: 1.18–3.57, p < 0.05) than among female participants. The participants who engaged in agriculture had significantly more *P. falciparum* malaria cases (OR = 3.47, 95% CI: 1.89–6.36, p < 0.001) than did the participants who engaged in nonagricultural activities.

**Table 1. Sociodemographic characteristics of the study participants (inhabitants of Walyte and Layignaw-Arguba villages), Gardula Zone, South Ethiopia (May-November 2018).**

| Characteristics | Number | Percentage (%) |
|---|---|---|
| **Number of households** | 329 | |
| **Number of participants** | 1672 | |
| **Age of participants** | | |
| Mean age | 22.8 (1–90) | |
| 1–4 year | 176 | 10.5 |
| 5–14 year | 446 | 26.7 |
| ≥15 year | 1050 | 62.8 |
| **Sex of participants** | | |
| Male | 833 | 49.8 |
| Female | 839 | 50.2 |
| **Educational level** | | |
| No education | 816 | 48.8 |
| Primary school | 738 | 44.2 |
| Above Primary | 118 | 7 |
| **Occupational level** | | |
| Agriculture | 779 | 46.6 |
| Government employ | 25 | 1.5 |
| Students | 579 | 34.6 |
| < 7 years* | 289 | 17.3 |

Key)
* = Preschool children

**Table 2. Incidence of malaria infection in Walyte and Layignaw-Arguba villages of Gardula Zone, South Ethiopia (May-November 2018).**

| Characteristics | Total examined (%) | *P. vivax* | | *P. falciparum* | |
|---|---|---|---|---|---|
| | | Incidence rate | OR (95% CI) | Incidence rate | OR (95% CI) |
| **Overall** | 4,884 | 0.06 | | 0.14 | |
| **Sex** | | | | | |
| Female | 50.5 | 0.04 | 1.00 | 0.09 | 1.00 |
| Male | 49.5 | 0.08 | 2.17 (0.93–5.04) | 0.19 | 2.05 (1.18–3.57) * |
| **Age** | | | | | |
| 1–4 years | 10.7 | 0.02 | 1.00 | 0.09 | 1.00 |
| 5–14 years | 25.8 | 0.08 | 3.34 (0.41–26.81) | 0.08 | 0.83 (0.24–2.77) |
| ≥ 15 years | 63.5 | 0.06 | 2.72 (0.36–20.55) | 0.17 | 1.92 (0.68–5.36) |
| **Occupation** | | | | | |
| Nonagriculture | 52.7 | 0.05 | 1.00 | 0.07 | 1.00 |
| Agriculture | 47.3 | 0.08 | 1.68 (0.75–3.74) | 0.22 | 3.47(1.89–6.36) * |
| **Education** | | | | | |
| No education | 49.7 | 0.07 | - | 0.18 | 2.66 (0.63–11.09) |
| Primary school | 43.2 | 0.06 | - | 0.10 | 1.48 (0.34–6.40) |
| Above primary | 7.1 | 0.00 | - | 0.07 | 1.00 |
| **Survey months** | | | | | |
| May | 34.2 | 0.09 | 4.02 (1.13–14.27) * | 0.20 | 3.14 (1.48–6.69) * |
| July | 34.2 | 0.02 | 1.00 | 0.06 | 1.00 |
| November | 31.6 | 0.08 | 3.63 (0.99–13.23) * | 0.16 | 2.43 (1.10–5.35) * |
| **Locality/village** | | | | | |
| Layignaw-Arguba | 47.4 | 0.05 | 1.00 | 0.07 | 1.00 |
| Walyte | 52.6 | 0.07 | 1.61 (0.71–3.64) | 0.21 | 3.09 (1.66–5.75) * |

Key)

* = statistically significant at P values < 0.05; 1.00 = reference group; OR = odds ratio and CI = confidence interval

## Malaria infection and travel history (univariate analysis)

Among the 4884 participants who tested for malaria at the three visits, 50.5% (2465) had traveled out of their permanent residence for at least one night in the last 30 days (Table 3). Most of them (78.9%, 1944) traveled to lowland malarious villages of Gardula Zone, namely, Argubatena, Gato, Holete and Wozeka. The majority of the travelers (82%, 2021) were in the ≥15 years age group, and most (61.3%, 1511) of them were involved in agricultural activities. Among the trips, 40.6% occurred from April to May (the planting season), and 35.2% occurred from October to November (the harvesting season).

Among the total malaria cases (82), 74.4% (61) were identified among the travelers. Among the 61 malaria infections detected among the travelers, 83.6% (51) traveled to malaria-endemic villages, and 80.3% (49) traveled for agricultural activities. Approximately 88.5% (54) of malaria infections detected among travelers were recorded during the planting (April and May) and harvesting (October and November) seasons. Approximately 85.2% (52) of travelers with malaria infection were individuals aged greater than or equal to 15 years. Most (70.5%, 43) malaria infections among travelers were recorded among nonusers of LLINs.

## Risk factors associated with malaria infection

Individuals who traveled away from their permanent residence in the past 30 days prior to the fingerprick (AOR = 2.60, 95% CI: 1.27–5.33) had a significantly greater risk of having *P*.

**Table 3. Travel history and malaria infection of inhabitants of Walyte and Layignaw-Arguba villages in Gardula Zone, South Ethiopia (May-November 2018).**

| Variables | Total, n = 4884 (%) | All positive, n = 82 (%) | *Pf, n = 57 (%)* | *Pv, n = 25 (%)* |
|---|---|---|---|---|
| **Travel history** | | | | |
| Yes | 2465 (50.5) | 61 (74.4) * | 45 (78.9) * | 16 (64.0) |
| No | 2419 (49.5) | 21 (25.6) | 12 (21.1) | 9 (36.0) |
| **Travel destination** | | | | |
| Lowland | 1944 (78.9) | 51 (83.6) | 37 (82.2) | 14 (87.5) |
| Highland | 521(21.1) | 10 (16.4) | 8 (17.8) | 2 (12.5) |
| **LLINs utilization during travel** | | | | |
| Yes | 862 (35.0) | 18 (29.5) | 13 (28.9) | 5 (31.2) |
| No | 1603 (65.0) | 43 (70.5) | 32 (71.1) | 11 (68.8) |
| **Purpose of travel** | | | | |
| Agriculture | 1511 (61.3) | 49 (80.3) | 38 (84.4) | 11 (68.8) |
| Other purposes | 954 (38.7) | 12 (19.7) | 7 (15.6) | 5 (31.2) |
| **Months travel most** | | | | |
| April -May | 1001 (40.6) | 31 (50.8) | 22 (48.9) | 9 (56.2) |
| June -July | 597 (24.2) | 7 (11.5) | 6 (13.3) | 1 (6.3) |
| October-November | 867 (35.2) | 23 (37.7) | 17 (37.8) | 6 (37.5) |
| **Locality/village** | | | | |
| Walyte | 1272 (51.6) | 43 (70.5) | 34 (75.6) | 9 (56.2) |
| Layignaw-Arguba | 1193 (48.4) | 18 (29.5) | 11 (24.4) | 7 (43.8) |
| **Age** | | | | |
| 1–4 years | 9 (0.4) | 1 (1.6) | 1 (2.2) | 0 (0.0) |
| 5–14 years | 435 (17.6) | 8 (13.2) | 4 (8.9) | 4 (25.0) |
| ≥ 15 years | 2021 (82.0) | 52 (85.2) | 40 (88.9) | 12 (75.0) |
| **Sex** | | | | |
| Male | 1484 (60.2) | 44 (72.1) | 32 (71.1) | 12 (75.0) |
| Female | 981 (39.8) | 17 (27.9) | 13 (28.9) | 4 (25.0) |

Key) Pf = P. falciparum; Pv = P. vivax and

* = statistically significant at the P value <0.05.

*falciparum* infection (Table 4). This was also true for all malaria cases (AOR = 2.01, 95% CI: 1.13–3.55). Males were 2.27 times more likely to be infected with malaria than females were (AOR = 2.27, 95% CI: 1.40–3.68). Those who reported agriculture as their main occupation were 2.05 times more likely to be infected with *P. falciparum* malaria than nonagricultural individuals were (AOR = 2.05, 95% CI: 1.00–4.17). Among the survey months, May and November had more malaria cases. The overall incidence of malaria infection was greater in Walyte than in Layignaw-Arguba (AOR = 3.05, 95% CI: 1.63–5.70).

## Bed net ownership and use

Two hundred eight, 63.2% of the study participants reported that their household had at least one LLIN (Table 5). Among those who reported owing LLINs, 52.9% (110) reported that at least one family member slept under bed net the night before the survey. Overall, 48.2% (53) of the LLIN users were mothers and children under five years of age, and 36.4% (40) were children under five years of age. Most of the LLIN owners (85.5%; 94) were found hanging in a sleeping place, and 81.8% (90) were in good condition (with no holes or physical damage). None of the surveyed households sprayed in the last 12 months.

**Table 4. Risk factors associated with *Plasmodium* parasite infection in Walyte and Layignaw-Arguba villages of Gardula Zone, South Ethiopia (May–November 2018).**

| Risk factor | *Pf*, AOR (95% CI) | *P value* | *Pv*, AOR (95% CI) | *P value* | All cases | *P value* |
|---|---|---|---|---|---|---|
| **Travel** | | | | | | |
| No | 1 | | 1 | | 1 | |
| Yes | 2.60* (1.27, 5.33) | 0.009 | 1.23 (0.53, 2.87) | 0.618 | 2.01* (1.13,3.55) | 0.016 |
| **Sex** | | | | | | |
| Female | 1 | | 1 | | 1 | |
| Male | 2.22* (1.25, 3.94) | 0.006 | 2.18 (0.94, 5.06) | 0.069 | 2.27* (1.40, 3.68) | 0.001 |
| **Age** | | | | | | |
| 1–4 years | - | | - | | 1 | |
| 5–14 years | - | | - | | 0.41 (0.09, 1.88) | 0.758 |
| ≥ 15 years | - | | - | | 1.19 (0.39. 3.64) | 0.254 |
| **Education** | | | | | | |
| No school | 3.13 (0.71,13.64) | 0.129 | - | | 4.38* (1.03,18.66) | 0.045 |
| Primary school | 1.72 (0.39,7.50) | 0.468 | - | | 2.77 (0.65,11.73) | 0.165 |
| Above primary | 1 | | - | | 1 | |
| **Occupation** | | | | | | |
| Nonagriculture | 1 | | - | | 1 | |
| Agriculture | 2.05* (1.00, 4.17) | 0.048 | - | | 1.83* (1.04, 3.23) | 0.036 |
| **Survey months** | | | | | | |
| May | 2.56* (1.18, 5.54) | 0.016 | 4.02* (1.13,14.29) | 0.031 | 2.92* (1.50, 5.67) | 0.002 |
| July | 1 | | 1 | | 1 | |
| November | 1.95 (0.87, 4.36) | 0.104 | 3.66* (1.00,13.34) | 0.049 | 2.33* (1.17, 4.64) | 0.015 |
| **Villages** | | | | | | |
| Layignaw-Arguba | 1 | | - | | 1 | |
| Walyte | 2.45* (1.49, 4.02) | <0.001 | - | | 3.05* (1.63, 5.70) | 0.001 |

Key) Pf = P. falciparum; Pv = P. vivax; AOR = adjusted odds ratio; CI = confidence interval;

* = statistically significant at P values <0.05 and 1.00 = reference group.

## Discussion

The findings of our study provide evidence of the possible spread of malaria to highland areas and the factors associated with malaria infection in highland dwellers. This study explored the incidence of malaria infection, the risk factors contributing to malaria transmission and the coverage and utilization of LLINs in Walyte and Layignaw-Arguba villages of Gardula Zone, South Ethiopia. These findings revealed that the habit of travel to malaria endemic lowland areas significantly increases the risk of malaria infection in the highland population of Gardula Zone. The risk of malaria infection was high among travelers who engaged in agricultural activities. This might be because the months of travel for agricultural activities coincide with peak malaria transmission seasons in lowland areas.

The current study revealed unexpectedly high malaria infection rates among the highland population of Gardula Zone. This finding is in line with other studies that indicated the

**Table 5. Bed net ownership and utilization in Walyte and Layignaw-Arguba villages of Gardula Zone, South Ethiopia (May–November 2018).**

| Characteristics | Walyte (HH = 171) | Layignaw-Arguba (HH = 158) | Total (HH = 329) |
|---|---|---|---|
| **Bed net ownership** | | | |
| Yes, N (%) | 105 (61.4) | 103 (65.2) | 208 (63.2) |
| No, N (%) | 66 (38.6) | 55 (34.8) | 121 (36.8) |
| **Number of LLINs per HH** | | | |
| 1 net, N (%) | 46 (43.8) | 75 (72.8) | 121 (58.2) |
| 2 nets, N (%) | 46 (43.8) | 24 (23.3) | 70 (33.6) |
| 3 and more net, N (%) | 13 (12.4) | 4 (3.9) | 17 (8.2) |
| **Source of bed net** | | | |
| Government, N (%) | 103 (98.0) | 100 (97.0) | 203 (97.6) |
| Self-purchased, N (%) | 2 (2.0) | 3 (3.0) | 5 (2.4) |
| **Any one slept under bed net last night?** | | | |
| Yes, N (%) | 56 (53.4) | 54 (52.4) | 110 (52.9) |
| No, N (%) | 49 (46.6) | 49 (47.6) | 98 (47.1) |
| **Who slept under the bed net?** | | | |
| Mother and children < 5 years, N (%) | 26 (46.4) | 27 (50.0) | 53 (48.2) |
| <5 years children only, N (%) | 15 (26.8) | 25 (46.3) | 40 (36.4) |
| Father and mother only, N (%) | 15 (26.8) | 2 (3.7) | 17 (15.4) |
| **Bed net hanged** | | | |
| Yes, N (%) | 45 (80.4) | 49 (90.7) | 94 (85.5) |
| No, N (%) | 11 (19.6) | 5 (9.3) | 16 (14.5) |
| **Bed net condition** | | | |
| No hole, N (%) | 48 (85.7) | 42 (77.7) | 90 (81.8) |
| Have hole, N (%) | 8 (14.3) | 12 (22.3) | 20 (18.2) |
| **Age of bed net** | | | |
| < 1 year, N (%) | 6 (5.7) | 8 (7.8) | 14 (6.7) |
| 1 year, N (%) | 43 (40.9) | 16 (15.5) | 59 (28.4) |
| 2 years, N (%) | 51 (48.6) | 77 (74.8) | 128 (61.5) |
| 3 years and more, N (%) | 5 (4.8) | 2 (1.9) | 7 (3.4) |

Key) HH = number of households

possible expansion of malaria transmission to highland areas, which are known to be free from local malaria transmission [8, 16, 21]. This expansion of malaria to highland areas was associated with climate change and population movement across different transects of malaria endemicity. On the other hand, the Ethiopian MIS of 2015 reported zero annual malaria incidences in the highlands of Ethiopia [3]. This might be due to the study design (a spot cross-sectional survey) and the small sample size for at least one study site that was followed via MIS, which could have led to the possibility of missing malaria infections. However, there is also a need to acknowledge the time gap between the last MIS, which was conducted in 2015, and the current study, which might be the reason for the increased incidence of malaria infection in the current study. Many malaria infections have been identified among nontravelers, including children aged less than five years, in the highlands of Gardula Zone. This finding, coupled with the detection of sporozoites from malaria vectors in previous studies [22, 23], might indicate the occurrence of local malaria transmission in the study area.

In this study, inhabitants who traveled to malaria endemic areas within the study period were more likely to be at risk of contracting *P. falciparum* malaria than those who did not

travel. This finding is consistent with other studies that reported malaria infection among the highland population of Ethiopia [17, 18, 26] and elsewhere [27–29]. The increased risk of malaria among travelers to endemic lowland areas might be associated with a lack of previous exposure and the associated immunity of people living in malaria-free highland villages. Thus, exposure of people with no or low immunity to malaria for an infected mosquito bite during their stay in lowlands can help them develop the disease more easily [17]. This might also be supported via the proportionally high nonuser rate of vector control interventions via highland travelers during their stay in malaria-endemic lowlands. The lack of access to and unsuitability of sleeping places at destinations might influence the nonuse of vector control interventions. Other studies also revealed an increased risk of malaria infection in traveling highland populations [17, 30, 31].

The current study documented varying levels of malaria infection between male and female participants. Similarly, studies conducted to address the impact of travel on malaria infection reported varying levels of malaria infection among male and female participants [26, 32–34]. The higher risk of malaria infection observed in males than in females could be explained by the fact that males are more likely to travel away from their permanent residence than females are [17].

We also revealed the varying malaria transmission risk across months of the year. More malaria cases were documented during May and November. This result is consistent with other studies performed in Butajira District, Ethiopia [35], and elsewhere [32, 36], in which temporal variation in malaria transmission has been documented. The months of May and November also coincide with the peak and minor malaria transmission seasons following long and short rainy seasons in the study area. We also observed that population movement to lowland areas for agricultural activities followed the rainy months.

In the present study, more malaria cases were documented in people living in or traveling from Walyte village. This finding is consistent with other studies performed in low-transmission areas of Ethiopia showing that malaria infection varies among villages [8]. Spatial variation has been documented in malaria transmission [32]. The possible explanations might be associated with the altitude (2100 masl), which is relatively lower. The variation in altitude and associated differences in temperature, rainfall and vegetation coverage might contribute to the varying malaria burdens. The low altitude of Walyte village might produce relatively high temperatures that might support vector and parasite development and malaria transmission. In addition, the documented high population movement in this village might incur high numbers of imported malaria cases, which serve as sources of infection for local malaria transmission.

We also reported that a lack of formal education was associated with an increased risk of infection with *Plasmodium* species. This result is consistent with other studies performed in Ethiopia [37] and sub-Saharan Africa [38]. This might be related to low levels of knowledge/ awareness about malaria transmission, prevention methods and treatment strategies. There were also more episodes of malaria among participants in the age group greater than or equal to 15 years. A similar result was documented in the highlands of the former Dirashe district, south Ethiopia, via a retrospective study [22]. This age group is mainly responsible for outdoor activities, including travel to lowland areas for different purposes, where they might acquire malaria infection.

Approximately 63.2% of households had at least one LLIN in the highlands of Gardula Zone. This shows that there has been improvement in bed net ownership since the last MIS, which reported that 33.7% of the surveyed households had at least one bed net in 2015 among highland dwellers. However, the utilization of bed nets has decreased since the last report of MIS. The bed net use rate was 52.9% of the households that owned at least one bed net in the

current study, which is slightly lower than the 57.1% reported for highland areas at the national level in 2015 [3].

One of the limitations of this study was the use of a low-sensitivity diagnostic tool, the RDT, for initial screening. RDT may miss infections with low parasite density and lead to a lower number of malaria cases than expected. The travel history of the participants was retrospectively assessed, which may have resulted in recall bias in the reporting of travel patterns. The other limitation of our study is the use of only epidemiological data of malaria to associate travel with malaria. However, to better understand the exact impact of travel on malaria risk, genomic analysis of Plasmodium parasites is important to determine the source of the parasite and whether it is locally transmitted or imported.

## Conclusions

The results of the present study revealed that the incidence of malaria infection in the highlands of Gardula Zone is unpredictably high and is associated mainly with the population traveling to malaria endemic villages in the same district. In addition, those participants who had no formal education and engaged in agricultural activities during travel were found to be at greater risk of developing malaria infection. The increased malaria infection in the highlands coupled with low bed net ownership and utilization might worsen the spread of malaria invasion to highland areas. Thus, malaria control and prevention programs should consider population movement between malaria-free highlands and malaria-endemic areas for effective malaria elimination.

## Acknowledgments

We would like to thank the former Dirashe district health office, the Agricultural office, the Health Center of Gidole, and the health posts of Layignaw-Arguba and Walyte villages for their cooperation, which provided useful and supporting information to conduct the study. The study participants, health extension workers and data collectors are highly acknowledged.

## Author Contributions

**Conceptualization:** Muluken Assefa, Fekadu Massebo, Teklu Wegayehu.

**Data curation:** Muluken Assefa, Fekadu Massebo, Teklu Wegayehu.

**Formal analysis:** Muluken Assefa, Temesgen Ashine, Teklu Wegayehu.

**Funding acquisition:** Fekadu Massebo, Teklu Wegayehu.

**Investigation:** Muluken Assefa, Teklu Wegayehu.

**Project administration:** Teklu Wegayehu.

**Software:** Muluken Assefa.

**Supervision:** Fekadu Massebo, Teklu Wegayehu.

**Writing – original draft:** Muluken Assefa, Temesgen Ashine.

**Writing – review & editing:** Fekadu Massebo, Teklu Wegayehu.

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
