## [Decision Letter · Decision Letter 0]

26 Aug 2024

PONE-D-24-07624Population travel increase the risk of Plasmodium falciparum infection in highland population of Dirashe District, Southwest Ethiopia: a longitudinal studyPLOS ONE

Dear Dr. Wegayehu,

Thank you for submitting your manuscript to PLOS ONE. After careful consideration, we feel that it has merit but does not fully meet PLOS ONE’s publication criteria as it currently stands. Therefore, we invite you to submit a revised version of the manuscript that addresses the points raised during the review process.

We look forward to receiving your revised manuscript.

Kind regards,

Musa Mohammed Ali, PhD

Academic Editor

PLOS ONE

“Arba Minch University.”

Reviewers' comments:

Reviewer's Responses to Questions

**Comments to the Author**

1. Is the manuscript technically sound, and do the data support the conclusions?

Reviewer #1: Partly

Reviewer #2: Yes

2. Has the statistical analysis been performed appropriately and rigorously? 

Reviewer #1: Yes

Reviewer #2: Yes

3. Have the authors made all data underlying the findings in their manuscript fully available?

Reviewer #1: Yes

Reviewer #2: Yes

4. Is the manuscript presented in an intelligible fashion and written in standard English?

Reviewer #1: No

Reviewer #2: Yes

5. Review Comments to the Author

Reviewer #1: Population travel increase the risk of Plasmodium falciparum infection in highland

population of Dirashe District, Southwest Ethiopia: a longitudinal study

This study's findings are highly intriguing and contribute significantly to understanding one of the potential causes of malaria spread in highlands, which are sometimes overlooked by nationwide targeted malaria control programs. The authors should update and rewrite the manuscript in a way that makes it legible and appealing to the audience.

Major Review.

1. Rewrite the discussion section and avoid repetition.

2. Evaluate the risk posed by travelers who do not own bed nets at home or who did not use bed nets but tested positive for Plasmodium infection upon return. This will also help to clarify how they can be contributing to local transmission after infection.

Minor reviews

1. Keywords

2. Revise first and second sentence in the introduction section for clarity (Page 10, Lines 43 – 47)

3. Theres no need of bracketing P. falciparum and P. vivax (Page 10, Lines 51-52)

4. This sentence need citation………Importantly, the increasing temperature may hasten the development of immature stages of malaria vectors, increasing the biting rates…..

5. This sentence is not clear……The non-immune populations traveling from the highlands to the lowlands for daily activities might increase the risk of getting malaria infection in the highland residents……

6. Rewrite….Moreover, local information on malaria incidence and associated risk factors of malaria transmission in highlands of Dirashe is patchy……

7. The sampling periods was chosen to coincided with a pick malaria transmission season following rainy seasons; and the dry season to measuring malaria infection……peak

8. Rewrite…………. In the laboratory, the slides were examined by microscope using recommended magnification power by senior laboratory technologist who was blinded for RDT results. The slide positive in first examination was considered as malaria infection. The second-round examination was conducted for negative slides in first round examination…….

9. Data were entered, edited…. What were the edits?

10. Tabulated age groups in the results section differs from what is reported in methodology section

11. …...(identified as P. falciparum and P. vivax infections)……Page 10..Line 194

12. Rewrite the following sentences…………..The incidence rates of P. falciparum and P. vivax has showed statistically significant variation among months in which the survey was conducted. While males had significantly higher P. falciparum malaria incidence rates (OR = 2.05, 95 % CI: 1.18-3.57, p < 0.05) than females, there was no sex difference in P. vivax malaria.

13. Rewrite the following sentences……………Most travelers 61.3% (1511) were travelled for agricultural activities. One thousand one, 40.6% of the participants were traveled during April to May (planting season), and 35.2% (867) were traveled during October to November (harvesting seasons). Of the total malaria cases (82), 74.4% (61) were identified among the travelers. Of all malaria cases among the travelers, majority 83.6% (51) were traveled to the malaria endemic villages; and 80.3% (49) were engaged in agriculture.

14. What could be the cause of the increased infection rates throughout the planting and harvest seasons? Are the infection patterns consistent between highlands and lowlands?

15. Results on bed net ownership versus infection level, and bed net use vs infection levels?

16. Those people engaged in agriculture had more risk of malaria infection since the population travel was related with agricultural activities……. Could agriculture be posing an increased risk of biting by malaria vectors?

17. Several evidences indicate the expansion of malaria transmission to highland areas which were known to be free from local malaria transmission……… What are the reasons for expansions as cited in the studies?

18. On the other hand, the Ethiopian MIS reported zero annual malaria incidences in the highlands of Ethiopia……Which year?

19. Majority of the travelers failed to use the vector control interventions while travelling to the lowlands….…. Which effective interventions are available to locals while travelling?

20. The move of highland dwellers, from low to high transmission settings makes them more susceptible to malaria infection, if exposed to an infective mosquito bite [15]… predispose them to getting an infectious bite from infected mosquitoes..

21. The higher risk of malaria infection observed in males as compared with females could be justified in that males are more likely travel away from their permanent residence than females for agricultural activities in the lowlands…..The sentence should be supported with data from the results of this study.

22. In addition, the travelers are less likely use the existing malaria control interventions due to lack of access and unsuitability of the sleeping places in the destination……cite relevant literature.

23. The traveler may stay there, and acquire Plasmodium parasites……… It is not certain whether the length of stay will enhance the likelihood of being bitten by an infected malaria vector.

24. About 63.2% of households had at least one LLINs in highlands of Dirashe District which is much higher than the report of Ethiopia MIS 2015 for highland areas above 2000m altitude which is 33.7% at national level………..Is there current report from Ethiopia MIS? If not, report this as an improvement from in LLIN ownership from 33.7% to 63.2%.

25. The bed net use rate was 52.9% of the households that owned at least one bed net, which, is slightly lower than 57.1% for highland areas at national level [3]…….. Why was the use rate lower with higher ownership?

Reviewer #2: The comments what I have for the Author are the following listed below

First of all this title by itself is great for the study area because malaria is now great problem in those area.

When I reviewd this manuscript its method is very clear and scientifically well written. There was no ethical issue problem also.

It is better if the Author re write conculsion what does the finding indicates and correct some reference

6. PLOS authors have the option to publish the peer review history of their article (what does this mean?). If published, this will include your full peer review and any attached files.

Reviewer #1: **Yes: **Kevin Omondi Ochwedo

Reviewer #2: **Yes: **Aberash Assefa Haile

---

## [Author Response · Author response to Decision Letter 0]

2 Oct 2024

First of all, we would like to extend our thankfulness to Editor’s and reviewers for their detailed appraisal. All the comments given are helpful and constructive. We accepted all the corrections, answered the questions, and tried to elaborate some sections. The amendments due to Editors’ and reviewers’ comments and major revisions we have made are highlighted by purple color. Language editions are also made at different sections. Major review such as rewrite the discussion section; and evaluate the risk posed by travelers are carefully considered. Point-by-point response and elaborations to the two reviewers are given below. 

Thank you!

//--------------------------

A) Reviewer #1:

Major review

Comment 1: Rewrite the discussion section and avoid repetition.

Response: The comment is well taken. We revised the discussion section accordingly. 

Comment 2: Evaluate the risk posed by travelers who do not own bed nets at home or who did not use bed nets but tested positive for Plasmodium infection upon return. This will also help to clarify how they can be contributing to local transmission after infection.

Response: The comment is well taken. We evaluated the risk reduced by bed net utilization among traveler. Despite the high number (n = 43) of cases were recorded among nonusers of the bed net, the proportion of malaria infection among nonusers (2.68) and users (2.09) of the bed net is not significantly different (table 3). The odd ratio of malaria infection among bed net user versus nonuser was 0.78, with the confidence interval that cross 1, 0.45-1.36. So, bed net use is not found to be protective from malaria infection among travelers. This might be due to non-suitability of the sleeping spaces to set the bed net in a way to prevent the contact between human host and the malaria vectors. 

Minor reviews

Comment 3: Keywords

Response: Thank you so much for your comment. It is corrected.

Comment 4: Revise first and second sentence in the introduction section for clarity

Response: The comment is well taken and amended as suggested (See introduction paragraph 1 and 2). 

Comment 5: There is no need of bracketing P. falciparum and P. vivax

Response: The comment is well taken and corrected accordingly (see introduction Para 2 line 4). 

Comment 6: This sentence need citation………Importantly, the increasing temperature may hasten the development of immature stages of malaria vectors, increasing the biting rates…..

Response: Thank you. We included reference in the revised version (Introduction Para 2 line 6).

Comment 7: This sentence is not clear……The non-immune populations traveling from the highlands to the lowlands for daily activities might increase the risk of getting malaria infection in the highland residents……

Response: The comment is also well taken. We revised the sentence (see introduction, Para 3 line 4).

Comment 8: Rewrite…. Moreover, local information on malaria incidence and associated risk factors of malaria transmission in highlands of Dirashe is patchy……

Response: Thank you. We revised the sentence (see introduction, Para, 5 line 6).

Comment 9: The sampling periods was chosen to coincided with a pick malaria transmission season following rainy seasons; and the dry season to measuring malaria infection……peak

Response: Thank you so much for your suggestion. We corrected accordingly (see method, Para 3, line 2).

Comment 10: Rewrite…………. In the laboratory, the slides were examined by microscope using recommended magnification power by senior laboratory technologist who was blinded for RDT results. The slide positive in first examination was considered as malaria infection. The second-round examination was conducted for negative slides in first round examination…….

Response: Thank you so much for your comments. The section was revised accordingly (see blood sample collection and processing section). 

Comment 11: Data were entered, edited…. What were the edits?

Response: Thank you so much for your constructive comments and suggestions. We miss spelled this. We revised this as “cleaned” (see data management and analysis, line 1). 

Comment 12: Tabulated age groups in the results section differs from what is reported in methodology section

Response: Thank you for your important comment. Yes, we lack consistence in reporting age categories, mainly in use of mathematical signs. We corrected this in the revised version of the manuscript (see data management and analysis (line 5) and the Tabulated age groups).

Comment 13: …... (identified as P. falciparum and P. vivax infections) ……Page 10...Line 194

Response: Thank you. This is to indicate the independent variable. We revised the sentence to make it clear (see Data management and analysis (para 2, line 6)).

Comment 14: Rewrite the following sentences…………. The incidence rates of P. falciparum and P. vivax has showed statistically significant variation among months in which the survey was conducted. While males had significantly higher P. falciparum malaria incidence rates (OR = 2.05, 95 % CI: 1.18-3.57, p < 0.05) than females, there was no sex difference in P. vivax malaria.

Response: Thank you for the comment. We rewritten the sentence to make it clear (see Malaria incidence, para 2, line 2).

Comment 15: 13. Rewrite the following sentences……………Most travelers 61.3% (1511) were travelled for agricultural activities. One thousand one, 40.6% of the participants were traveled during April to May (planting season), and 35.2% (867) were traveled during October to November (harvesting seasons). Of the total malaria cases (82), 74.4% (61) were identified among the travelers. Of all malaria cases among the travelers, majority 83.6% (51) were traveled to the malaria endemic villages; and 80.3% (49) were engaged in agriculture.

Response: Thank you. We rewritten the sentence to make it clear (Malaria infection and travel history, line 6).

Comment 16: What could be the cause of the increased infection rates throughout the planting and harvest seasons? Are the infection patterns consistent between highlands and lowlands?

Response: Thank you so much for your question. The plausible reason might be, both the planting and harvesting seasons were coincide with the peak malaria transmission seasons in the area.

Comment 17: Results on bed net ownership versus infection level, and bed net use vs infection levels?

Response: Thank you for important question. As the response provided for “comment 2”, bed net utilization during travel did not come out as protective among travelers.

Comment 18: Those people engaged in agriculture had more risk of malaria infection since the population travel was related with agricultural activities……. Could agriculture be posing an increased risk of biting by malaria vectors?

Response: Thank you for your important argument. The authors not believe that engagement on the agriculture activity by itself expose for malaria infection. And also, usually agricultural activities are cared out during the day time, the possibility of get mosquito bite is less by being agricultural worker. The increase in malaria infection among participants engaged in agricultural activity might be due to 1) the sleeping space that they stay over night is not suitable to create complete barrier between human host and the mosquitoes, thus the mosquitoes can get access to human host overnight, 2) most of agricultural worker are illiterate, might have low knowledge on the malaria transmission and how to prevent the disease, 3) most importantly, the migrant worker were not taken as the target population by any malaria control and prevention program in the study area, so, lack of access for awareness development education and bed net might played the major role in increased malaria infection among agricultural workers. As we also indicate in our discussion, the utilization of bed net among travelers was low.

Comment 19: Several evidences indicate the expansion of malaria transmission to highland areas which were known to be free from local malaria transmission……… What are the reasons for expansions as cited in the studies?

Response: Thank you for your question. We included the plausible explanation for malaria expansion to the highland areas in the revised version of the manuscript (See discussion para 2 line 4).

Comment 20: On the other hand, the Ethiopian MIS reported zero annual malaria incidences in the highlands of Ethiopia……Which year?

Response: Thank you for the important comment. We revised the section to include the date of the last MIS that conducted in 2015 (See discussion, para 2, line 7)

Comment 21: Majority of the travelers failed to use the vector control interventions while travelling to the lowlands….…. Which effective interventions are available to locals while travelling?

Response: Thank you for your important question. Actually, there is no active malaria control program at the travel destine in the current study setting. The travelers need to carry the bed net that they provided at there home village to use during travel. This is one of the reasons for low utilization of bed net among travelers (See discussion, para 3, line 12).

Comment 22: The move of highland dwellers, from low to high transmission settings makes them more susceptible to malaria infection, if exposed to an infective mosquito bite [15] … predispose them to getting an infectious bite from infected mosquitoes.

Response: Thank you for your important suggestion. Revised accordingly (See discussion, para 3, line 12-19).

Comment 23: The higher risk of malaria infection observed in males as compared with females could be justified in that males are more likely travel away from their permanent residence than females for agricultural activities in the lowlands…. The sentence should be supported with data from the results of this study.

Response: Thank you. This is from simple observation during the study period and long-standing culture of the study area that male are responsible for agricultural activities.

Comment 24: In addition, the travelers are less likely use the existing malaria control interventions due to lack of access and unsuitability of the sleeping places in the destination……cite relevant literature.

Response: Thank you for the important comment. This is from our finding (Table 3).

Comment 25: The traveler may stay there, and acquire Plasmodium parasites……… It is not certain whether the length of stay will enhance the likelihood of being bitten by an infected malaria vector.

Response: Thank you for your important argument. The sentence is revised to make it clear (See discussion, para 8, line 6).

Comment 26: About 63.2% of households had at least one LLINs in highlands of Dirashe District which is much higher than the report of Ethiopia MIS 2015 for highland areas above 2000m altitude which is 33.7% at national level………. Is there current report from Ethiopia MIS? If not, report this as an improvement from in LLIN ownership from 33.7% to 63.2%.

Response: Thank you for the comment. The last MIS was conducted in 2015. There was no other report since then. We corrected the statement accordingly (See discussion, para 9, line 1).

Comment 27: The bed net use rate was 52.9% of the households that owned at least one bed net, which, is slightly lower than 57.1% for highland areas at national level [3] ……. Why was the use rate lower with higher ownership?

Response: Thank you for the question. The authors believe that even there is difference in the figures of utilization of bed net, it is not significant and within expected limit of variation that might occur due to different reasons. The difference in the study area, study time period, sample size etc.

B) Reviewer #2:

The comments what I have for the Author are the following listed below. First of all this title by itself is great for the study area because malaria is now great problem in those area. When I reviewing this manuscript, its method is very clear and scientifically well written. There was no ethical issue problem also.

Comment 1: It is better if the Author re write conclusion what does the finding indicates

Response: Thank you so much for your good suggestion. We corrected accordingly (See the conclusion).

Comment 2: Correct some reference

Response: Thank you so much for your interesting comments. We corrected accordingly (See the reference).

---

## [Decision Letter · Decision Letter 1]

4 Nov 2024

PONE-D-24-07624R1Population travel increases the risk of Plasmodium falciparum infection in the highland population of Dirashe District, Southwest Ethiopia: a longitudinal studyPLOS ONE

Dear Dr. Wegayehu,

Thank you for submitting your manuscript to PLOS ONE. After careful consideration, we feel that it has merit but does not fully meet PLOS ONE’s publication criteria as it currently stands. Therefore, we invite you to submit a revised version of the manuscript that addresses the points raised during the review process.

Academic editor: 

Figure 1 appears to represent Ethiopia and Southern Ethiopia as two separate countries. Please revise the figures accordingly. Additionally, instead of using the term "illiterate," I recommend using "participants with no formal education," as the translation of "illiterate" into the local language may be considered offensive.

We look forward to receiving your revised manuscript.

Kind regards,

Musa Mohammed Ali, PhD

Academic Editor

PLOS ONE

Journal Requirements:

Reviewers' comments:

Reviewer's Responses to Questions

**Comments to the Author**

1. If the authors have adequately addressed your comments raised in a previous round of review and you feel that this manuscript is now acceptable for publication, you may indicate that here to bypass the “Comments to the Author” section, enter your conflict of interest statement in the “Confidential to Editor” section, and submit your "Accept" recommendation.

Reviewer #1: All comments have been addressed

Reviewer #2: All comments have been addressed

2. Is the manuscript technically sound, and do the data support the conclusions?

Reviewer #1: Yes

Reviewer #2: Yes

3. Has the statistical analysis been performed appropriately and rigorously? 

Reviewer #1: Yes

Reviewer #2: Yes

4. Have the authors made all data underlying the findings in their manuscript fully available?

Reviewer #1: Yes

Reviewer #2: Yes

5. Is the manuscript presented in an intelligible fashion and written in standard English?

Reviewer #1: Yes

Reviewer #2: Yes

6. Review Comments to the Author

Reviewer #1: (No Response)

Reviewer #2: In my point of view their is no any ethical problem for this article, in addition to this l haven't seen competing interest between authors, there is no ethical problem among reach ethic as well as publication . l hope this publication add some value for the community

7. PLOS authors have the option to publish the peer review history of their article (what does this mean?). If published, this will include your full peer review and any attached files.

Reviewer #1: **Yes: **Kevin Omondi Ochwedo

Reviewer #2: **Yes: **Aberash Assefa Haile

---

## [Author Response · Author response to Decision Letter 1]

30 Nov 2024

Response to Reviewers

First of all, we would like to extend our thankfulness to Editor’s and reviewers for their detailed appraisal. A response that addresses the points raised during the review process are given below. 

Thank you!

//--------------------------

A) Academic editor: 

Figure 1 appears to represent Ethiopia and Southern Ethiopia as two separate countries. Please revise the figures accordingly. 

Response: The comment is well taken and Figure 1 is revised. Accordingly, the caption is also re-narrated (See Figure 1 (separate file), and the caption on page 6 in the manuscript)

Additionally, instead of using the term "illiterate," I recommend using "participants with no formal education," as the translation of "illiterate" into the local language may be considered offensive.:

Response: The comment is well taken and amended accordingly (See pages 2 and 22).

Journal Requirement

Response: The reference list is reviewed to ensure its completeness and correctness. Accordingly, corrections have been made (see pages 25-30).

Journal requirement

The previous submission has no line numbers. Now, the line numbers are inserted in the manuscript file.

Formatting issues: line spacing has been checked and corrected throughout the document 

B) Reviewers: The reviewers were satisfied with the previous response given. It can be seen from their response.

C) Other corrections

Ethiopia has several administrative regions. This study was conducted in former South Nations, Nationalities and People’s Regional States (SNNPRs). Of course, it is political boundary. Currently the region is separated into four administrative regions. The region in which this study was conducted is found in South Ethiopia. To indicate this geographical location the phrase ‘Southwest Ethiopia’ is replaced by ‘South Ethiopia’ in the MS (see pages 1.2, 5,6, Figure Tables). 

The study district ‘Dirashe district’ is currently named as ‘Gardula Zone’ with the same geographical location following the division of the former SNNPRs. Still, it is an administrative boundary. To indicate this name change with the same geographical area, the phrase ‘Dirashe district’ is replaced by ‘Gardula Zone’ in the MS (see the Title, Tables and sections of the MS).

---

## [Editor Report · Decision Letter 2]

4 Dec 2024

Population travel increases the risk of Plasmodium falciparum infection in the highland population of Gardula Zone, South Ethiopia: a longitudinal study

PONE-D-24-07624R2

Dear Dr. Wegayehu,

We’re pleased to inform you that your manuscript has been judged scientifically suitable for publication and will be formally accepted for publication once it meets all outstanding technical requirements.

Kind regards,

Musa Mohammed Ali, PhD

Academic Editor

PLOS ONE
---

## [Editor Report · Acceptance letter]

10 Dec 2024

PONE-D-24-07624R2 

PLOS ONE

Dear Dr. Wegayehu, 

I'm pleased to inform you that your manuscript has been deemed suitable for publication in PLOS ONE. Congratulations! Your manuscript is now being handed over to our production team.

Kind regards, 

on behalf of

Dr. Musa Mohammed Ali 

Academic Editor

PLOS ONE